# K-ADAPTER: INFUSING KNOWLEDGE INTO PRE-TRAINED MODELS WITH ADAPTERS

## ABSTRACT

We study the problem of injecting knowledge into large pre-trained models like BERT and RoBERTa. Existing methods typically update the original parameters of pre-trained models when injecting knowledge. However, when multiple kinds of knowledge are injected, they may suffer from catastrophic forgetting. To address this, we propose K-ADAPTER, which retains the original parameters of the pre-trained model fixed and supports continual knowledge infusion. Taking RoBERTa as the pre-trained model, K-ADAPTER has a neural adapter for each kind of infused knowledge, like a plug-in connected to RoBERTa. There is no information flow between different adapters, thus different adapters are efficiently trained in a distributed way. We inject two kinds of knowledge, including factual knowledge obtained from automatically aligned text-triplets on Wikipedia and Wikidata, and linguistic knowledge obtained from dependency parsing. Results on three knowledge-driven tasks (total six datasets) including relation classification, entity typing and question answering demonstrate that each adapter improves the performance, and the combination of both adapters brings further improvements. Probing experiments further indicate that K-ADAPTER captures richer factual and commonsense knowledge than RoBERTa.

## 1 INTRODUCTION

Language representation models, which are pre-trained on large-scale text corpus through unsupervised objectives like (masked) language modeling, such as BERT (Devlin et al., 2019), GPT (Radford et al., 2018; 2019), XLNet (Yang et al., 2019), RoBERTa (Liu et al., 2019) and T5 (Raffel et al., 2019), have established state-of-the-art performances on various NLP downstream tasks.

Despite the huge success of these pre-trained models in empirical studies, recent studies suggest that models learned in such an unsupervised manner struggle to capture rich knowledge. For example, Poerner et al. (2019) suggest that although language models do well in reasoning about the surface form of entity names, they fail in capturing rich factual knowledge. Kassner & Schütze (2019) observe that BERT mostly did not learn the meaning of negation (e.g. "not"). These observations motivate us to study the injection of knowledge into pre-trained models like BERT and RoBERTa.

Recently, some efforts have been made to exploit injecting knowledge into pre-trained language models (Zhang et al., 2019; Lauscher et al., 2019; Levine et al., 2019; Peters et al., 2019; He et al., 2019; Xiong et al., 2020). Most previous works (as shown in Table 1) augment the standard language modeling objective with knowledge-driven objectives and update model parameters in a multi-task learning manner. Although these methods, with updated pre-trained models, obtain better performance on downstream tasks, they fail at continual learning (Kirkpatrick et al., 2017). Model parameters need to be retrained when new kinds of knowledge are injected, which may result in the catastrophic forgetting of previously injected knowledge. Meanwhile, the resulting pre-trained models produce entangled representations, which makes it hard to investigate the effect of each knowledge when multiple kinds of knowledge are injected.

In this paper, we propose K-ADAPTER, a flexible and simple approach that infuses knowledge into large pre-trained models. K-ADAPTER has attractive properties including supporting continual knowledge infusion and producing disentangled representations. It leaves the original representation of a pre-trained model unchanged and exports different representations for different types of infused knowledge. This is achieved by the integration of compact neural models, dubbed adapters here.

Table 1: Comparison between our approach (K-ADAPTER) and previous works on injecting knowledge into BERT.

| Model | Knowledge Source | Objective | BERT fixed in training? | Continual knowledge infusion? |
|---|---|---|---|---|
| ERNIE (Zhang et al., 2019) | Wikipedia, WikiData | entity linking | N | N |
| LIBERT (Lauscher et al., 2019) | WordNet | synonym word prediction, hyponym-hypernym prediction | from scratch | N |
| SenseBERT (Levine et al., 2019) | WordNet | word-supersense prediction | from scratch | N |
| KnowBERT (Peters et al., 2019) | Wordnet, Wikipedia, CrossWikis | entity linking , hypernym linking | N | N |
| WKLM (Xiong et al., 2020) | WikiPedia, WikiData | replaced entity detection | N | N |
| BERT-MK (He et al., 2019) | Unified Medical Language System | discriminate between real and fake facts | N | N |
| K-Adapter (this work) | Wikipedia, Wikidata, dependency parser | predication prediction, dependency relation prediction | Y | Y |

Adapters are knowledge-specific models plugged outside of a pre-trained model, whose inputs are the output hidden-states of intermediate layers of the pre-trained model. We take RoBERTa (Liu et al., 2019) as the base pre-trained model and integrate two types of knowledge, including factual knowledge obtained by aligned Wikipedia text to Wikidata triplets, linguistic knowledge obtained by applying off-the-shell dependency parser to web texts. In the pre-training phase, we train two adapters independently on relation classification task and dependency relation prediction task respectively, while keeping the original parameters of RoBERTa frozen. Since adapters have much less trainable parameters compared with RoBERTa, the training process is memory efficient.

We conduct extensive experiments on six benchmark datasets across three knowledge-driven tasks, i.e., relation classification, entity typing and question answering. Experiments show that K-ADAPTER consistently performs better than RoBERTa, and achieves state-of-the-art performance on five datasets, and comparable performance compared with CosmosQA SOTA. Probing experiments on LAMA (Poerner et al., 2019) and LAMA-UHN (Petroni et al., 2019), further demonstrates that K-ADAPTER captures richer factual and commonsense knowledge than RoBERTa.

The contributions of this paper are summarized as follows:

- We propose K-ADAPTER, a flexible approach that supports continual knowledge infusion into large pre-trained models (e.g. RoBERTa in this work).

- We infuse factual knowledge and linguistic knowledge, and show that adapters for both kinds of knowledge work well on downstream tasks.

- K-ADAPTER achieves superior performance by fine-tuning parameters on three downstream tasks, and captures richer factual and commonsense knowledge than RoBERTa on probing experiments.

## 2 RELATED WORK

Our work relates to the area of injecting knowledge into pre-trained models. As stated in Table 1, previous works mainly differ from the knowledge sources and the objective used for training.

**ERNIE** (Zhang et al., 2019) injects a knowledge graph into BERT. They align entities from Wikipedia sentences to fact triples in WikiData, and discard sentences with less than three entities. In the training process, the input includes sentences and linked facts, and the knowledge-aware learning objective is to predict the correct token-entity alignment. Entity embeddings are trained on fact triples from WikiData via TransE (Bordes et al., 2013). **LIBERT** (Lauscher et al., 2019) injects pairs of words with synonym and hyponym-hypernym relations in WordNet. The model takes a pair of words separated by a special token as the input, and is optimized by a binary classification problem, which predicts whether the input holds a particular relation or not. **SenseBERT** (Levine et al., 2019) considers word-supersense knowledge. It inject knowledge by predicting the supersense of the masked word in the input, where the candidates are nouns and verbs and the ground truth comes

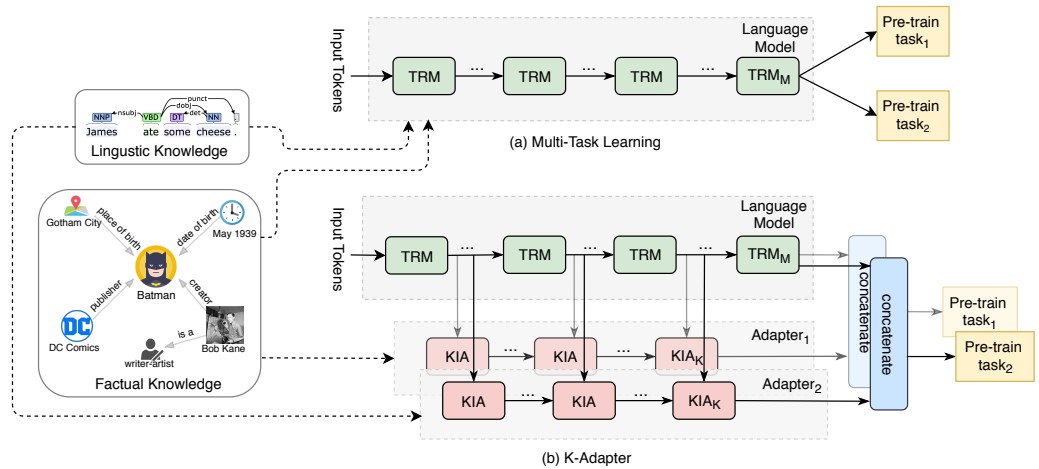

Figure 1: (a) Pre-trained language models inject multiple kinds of knowledge with multi-task learning. Model parameters need to be retrained when injecting new kinds of knowledge, which may result in the catastrophic forgetting (b) Our K-ADAPTER injects multiple kinds of knowledge by training adapters independently on different pre-train tasks, which supports continual knowledge infusion. When we inject new kinds of knowledge, the existing knowledge-specific adapters will not be affected. KIA represents the adapter layer and TRM represents the transformer layer, both of which are shown in Figure 2.

from WordNet. **KnowBERT** (Peters et al., 2019) incorporates knowledge bases into BERT using Knowledge attention and recontextualization, where the knowledge comes from synset-synset and lemma-lemma relationships in WordNet, and entity linking information in Wikipedia. If entity linking supervision is available, the model is learned with an additional knowledge-aware log-likelihood or max-margin objective. **WKLM** (Xiong et al., 2020) replaces entity mentions in the original document with names of other entities of the same type. The model is trained to distinguish the correct entity mention from randomly chosen ones. **BERT-MK** (He et al., 2019) integrates fact triples from knowledge graph. For each entity, it sample incoming and outcoming instances from the neighbors on the knowledge graph, and replaces head or tail entity to create negative instances. The model is learned to discriminate between real and fake facts.

As shown in Table 1, our model (K-ADAPTER) differs from previous studies in three aspects. First, we consider both fact-related objective (i.e. predicate/relation prediction) and linguistic-related objective (i.e. dependency relation prediction). Second, the original parameter of BERT is clamped in the knowledge infusion process. Third, our approach supports continual learning, which means that the learning of different adapters are not entangled. This flexibility enables us to efficiently inject different types of knowledge independently, and inject more types of knowledge without any loss on the previously injected knowledge.

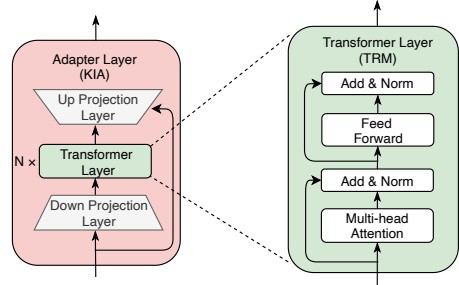

Figure 2: Structure of the adapter layer (left). The adapter layer consists of two projection layers and $N=2$ transformer layers, and a skip-connection between two projection layers.

## 3 K-ADAPTER

As illustrated in Figure 1 (a), most of the previous works enhance pre-trained language models by injecting knowledge and update model parameters through multi-task learning. Regardless of these different versions of knowledge-injected methods with multi-task learning, common issues not fully studied are catastrophic forgetting of previous knowledge. To address this, we present K-ADAPTER as shown in Figure 1(b), where multiple kinds of knowledge are injected into different compact neural models (i.e., adapters in this paper) individually instead of directly injecting knowledge into pre-trained models. It keeps the original representation of a pre-trained model fixed and

supports continual knowledge infusion, i.e., injecting each kind of knowledge into the corresponding knowledge-specific adapter and producing disentangled representation. Specifically, adapters are knowledge-specific models (with few parameters) plugged outside of a pre-trained model. The inputs of adapters are the output hidden-states of intermediate layers of the pre-trained model. Each adapter is pre-trained independently on different tasks for injecting discriminative knowledge while the original parameters of the pre-trained model are frozen. In this paper, we exploit RoBERTa (Liu et al., 2019) as the pre-trained model, and mainly infuse factual knowledge and linguistic knowledge with two kinds of adapters, i.e., factual adapter and linguistic adapter which are pre-trained on the relation classification task and dependency relation prediction task respectively. In this section, we first describe the structure of our adapter, and then present the process of pre-training knowledge-specific adapters.

## 3.1 ADAPTER STRUCTURE

In this work, we present a different adapter structure as shown in Figure 2, which is referred to as the knowledge-specific adapter. In contrast to Houlsby et al. (2019) add adapter layers into each transformer layer, our adapter works as outside plug-ins. Each adapter model consists of $K$ adapter layers that contain $N$ transformer (Vaswani et al., 2017) layers and two projection layers. A skip-connection is applied across two projection layers. Specifically, for each adapter model, we plug adapter layers among different transformer layers of the pre-trained model. We concatenate the output hidden feature of the transformer layer in the pre-trained model and the output feature of the former adapter layer, as the input feature of the current adapter layer. For each knowledge-specific adapter, we concatenate the last hidden features of the pre-trained model and adapter as the final output feature of this adapter model.

In the pre-training procedure, we train each knowledge-specific adapter on different pre-training tasks individually. For various downstream tasks, K-ADAPTER can adopt the fine-tuning procedure similar to RoBERTa and BERT. When only one knowledge-specific adapter is adopted, we can take the final output feature of this adapter model as the input for task-specific layers of the downstream task. When multiple knowledge-specific adapters are adopted, we concatenate the output features of different adapter models as the input for task-specific layers of the downstream task.

## 3.2 PRE-TRAINING SETTINGS

We use RoBERTa$_{LARGE}$ (L=24, H=1024, A=16, 355M params) implementation by Huggingface[1] as the pre-trained model in all our experiments. As for each adapter layer, we denote the number of transformer layer as $N$, the hidden dimension of transformer layer as $H_A$, the number of self-attention heads as $A_A$, the hidden dimension of down-projection and up-projection layers as $H_d$ and $H_u$. In detail, we have the following adapter size: $N = 2$, $H_A = 768$, $A_A = 12$, $H_u = 1024$ and $H_d = 768$. The RoBERTa layers where adapter layers plug in are {0,11,23}, and different adapter layers do not share parameters. Thus the total parameters for each adapter model are about 42M, which are much smaller than RoBERTa$_{LARGE}$ and make the training process memory efficient. It should be noticed that RoBERTa is fixed during training and the parameters of adapters are trainable and initialized randomly. Then we describe how to inject different knowledge into knowledge-specific adapters as below.

## 3.3 FACTUAL ADAPTER

Factual knowledge can be described as the basic information that is concerned with facts. In this work, we acquire factual knowledge from the relationships among entities in natural language. We extract a sub-dataset T-REx-rc from T-REx (ElSahar et al., 2018) which is a large scale alignment dataset between Wikipedia abstracts and Wikidata triples. We discard all relations having less than 50 entity pairs, collecting 430 relations and 5.5M sentences. In order to inject factual knowledge, we propose to pre-train a knowledge-specific adapter called facAdapter on the relation classification task. This task requires a model to classify relation labels of given entity pairs based on context. Specifically, the last hidden features of RoBERTa and facAdapter are concatenated as the input representation, and the pooling layer is applied to the input representations of the given entities. Then, we concatenate two entity representations to perform relation classification.

---

[1]https://github.com/huggingface/transformers

Table 2: Results on two entity typing datasets OpenEntity and FIGER.

| Model | OpenEntity | | | FIGER | | |
|---|---|---|---|---|---|---|
| | P | R | Mi-$F_1$ | Acc | Ma-$F_1$ | Mi-$F_1$ |
| NFGEC (Shimaoka et al., 2016) | 68.80 | 53.30 | 60.10 | 55.60 | 75.15 | 71.73 |
| BERT-base (Zhang et al., 2019) | 76.37 | 70.96 | 73.56 | 52.04 | 75.16 | 71.63 |
| ERNIE (Zhang et al., 2019) | 78.42 | 72.90 | 75.56 | 57.19 | 75.61 | 73.39 |
| KnowBERT (Peters et al., 2019) | 78.60 | 73.70 | 76.10 | - | - | - |
| KEPLER (Wang et al., 2019) | 77.20 | 74.20 | 75.70 | - | - | - |
| WKLM (Xiong et al., 2020) | - | - | - | 60.21 | 81.99 | 77.00 |
| RoBERTa | 77.55 | 74.95 | 76.23 | 56.31 | 82.43 | 77.83 |
| RoBERTa + multitask | 77.96 | 76.00 | 76.97 | 59.86 | 84.45 | 78.84 |
| K-ADAPTER (F+L) | 78.99 | 76.27 | **77.61** | 61.81 | 84.87 | **80.54** |
| K-ADAPTER (F) | 79.30 | 75.84 | 77.53 | 59.50 | 84.52 | 80.42 |
| K-ADAPTER (L) | 80.01 | 74.00 | 76.89 | 61.10 | 83.61 | 79.18 |
| K-ADAPTER (w/o knowledge) | 74.47 | 74.91 | 76.17 | 56.93 | 82.56 | 77.90 |

## 3.4 LINGUISTIC ADAPTER

Linguistic knowledge is implicitly contained in natural language texts, e.g., syntactic and semantic information. In this work, we acquire linguistic knowledge from dependency relationships among words in natural language text. We build a dataset consisting of 1M examples. In particular, we run the off-the-shell dependency parser from Stanford Parser[2] on a part of Book Corpus (Zhu et al., 2015). To inject linguistic knowledge, we pre-train another knowledge-specific adapter called linAdapter on the task of dependency relation prediction. This task aims to predict the head index of each token in the given sentence. We concatenate the last hidden features of RoBERTa and linAdapter as the input representation, and then apply a linear layer to input representations of each token to perform classification. More training details of facAdapter and linAdapter can be found in the supplementary material.

## 4 EXPERIMENTS

We evaluate our K-ADAPTER on three knowledge-driven downstream tasks, i.e., entity typing, question answering and relation classification. Furthermore, we conduct probing experiments to examine the ability of models for learning factual knowledge. The notations of K-ADAPTER (F+L), K-ADAPTER (F), and K-ADAPTER (L) denote our model which consists of both factual adapter and linguistic adapter, only factual adapter and only linguistic adapter, respectively. The implementation details, and statistics of datasets are in the supplementary material.

## 4.1 ENTITY TYPING

We conduct experiments on fine-grained entity typing which aims to predict the types of a given entity and its context. We evaluate our models on OpenEntity (Choi et al., 2018) and FIGER (Ling et al., 2015) following the same split setting as Zhang et al. (2019). To fine-tune our models for entity typing, we modify the input token sequence by adding the special token "@" before and after a certain entity, then the first "@" special token representation is adopted to perform classification. As for OpenEntity, we adopt micro $F_1$ score as the final metric to represent the model performance. As for FIGER, we adopt strict accuracy, loose macro, loose micro $F_1$ scores (Ling & Weld, 2012) for evaluation following the same evaluation criteria used in previous works.

**Baselines** **NFGEC (Shimaoka et al., 2016)** employs attentive recursive neural networks to compose context representations. **KEPLER (Wang et al., 2019)** integrates factual knowledge with the supervision of the knowledge embedding objective. **RoBERTa+multitask** is our RoBERTa model pre-trained with multi-task learning (as shown in Figure 1(a)) for injecting multiple kinds of knowledge on two pre-training tasks. **K-ADAPTER (w/o knowledge)** consists of a RoBERTa model and an adapter without being injected knowledge. Other baseline models, such as BERT-base, ERNIE, KnowBERT and WKLM are described in Section 2.

---

[2]http://nlp.stanford.edu/software/lex-parser.html

Table 3: Results on question answering datasets including: CosmosQA, SearchQA and Quasar-T.

| Model | SearchQA | | Quasar-T | | CosmosQA |
|---|---|---|---|---|---|
| | EM | $F_1$ | EM | $F_1$ | Accuracy |
| BiDAF (Seo et al., 2016) | 28.60 | 34.60 | 25.90 | 28.50 | - |
| AQA (Buck et al., 2018) | 40.50 | 47.40 | - | - | - |
| R^3 (Wang et al., 2017a) | 49.00 | 55.30 | 35.30 | 41.70 | - |
| DSQA (Lin et al., 2018) | 49.00 | 55.30 | 42.30 | 49.30 | - |
| Evidence Agg. (Wang et al., 2018) | 57.00 | 63.20 | 42.30 | 49.60 | - |
| BERT (Xiong et al., 2020) | 57.10 | 61.90 | 40.40 | 46.10 | - |
| WKLM (Xiong et al., 2020) | 58.70 | 63.30 | 43.70 | 49.90 | - |
| WKLM + Ranking (Xiong et al., 2020) | 61.70 | 66.70 | 45.80 | 52.20 | - |
| BERT-FT$_{RACE+SWAG}$ (Huang et al., 2019) | - | - | - | - | 68.70 |
| RoBERTa | 59.01 | 65.62 | 40.83 | 48.84 | 80.59 |
| RoBERTa + multitask | 59.92 | 66.67 | 44.62 | 51.17 | 81.19 |
| K-ADAPTER (F+L) | **61.96** | **67.31** | **46.32** | **53.00** | **81.83** |
| K-ADAPTER (F) | 61.85 | 67.17 | 46.20 | 52.86 | 80.93 |
| K-ADAPTER (L) | 61.15 | 66.82 | 45.66 | 52.39 | 80.76 |

**Results and Discussion**     The results on OpenEntity and FIGER are shown in Table 2. K-ADAPTER (F+L) achieves consistent improvements across these datasets. As for OpenEntity, our RoBERTa achieve better results than other baseline models. K-ADAPTER (F+L) further achieves improvement of 1.38% $F_1$ over RoBERTa, which means factual knowledge and linguistic knowledge help to predict the types more accurately. As for FIGER, it covers more entity types, and is more fine-grained than OpenEntity. Compared with WKLM, K-ADAPTER (F+L) improves the macro $F_1$ by 2.88%, micro $F_1$ by 2.54% and accuracy by 1.60%. This demonstrates that K-ADAPTER (F+L) benefits fine-grained entity typing. In addition, we further conduct several experiments on our ablated model K-ADAPTER (w/o knowledge), to explore whether the performance gains came from introducing knowledge or additional parameters. Results show that K-ADAPTER (F) significantly outperforms K-ADAPTER (w/o knowledge). Moreover, it is worth noting that on OpenEntity dataset, K-ADAPTER (w/o knowledge) even performs slightly worse than RoBERTa. These results demonstrate that our model gains improvement from knowledge instead of more parameters. Thus, for simplicity, we don't discuss K-ADAPTER (w/o knowledge) in the following experiments.

## 4.2 QUESTION ANSWERING

We conduct experiments on two question answering (QA) tasks, i.e., commonsense QA and open-domain QA. Commonsense QA aims to answer questions with commonsense. We adopt CosmosQA (Huang et al., 2019) to evaluate our models. CosmosQA requires commonsense-based reading comprehension, formulated as multiple-choice questions. To fine-tune our models for CosmosQA, the input token sequence is modified as *"<SEP>context </SEP>question</SEP>answer</SEP>"*, then the representation of the first token is adopted to perform classification, and will get a score for this answer. After getting four scores, the answer with the highest score will be selected. We report accuracy scores obtained from the leaderboard.

Open-domain QA aims to answer questions using external resources such as collections of documents and webpages. We evaluate our modes on two public datasets, i.e., Quasar-T (Dhingra et al., 2017) and SearchQA (Dunn et al., 2017). Specifically, we first retrieve paragraphs corresponding to the question using the information retrieval system and then extract the answer from these retrieved paragraphs through the reading comprehension technique. Following previous work(Lin et al., 2018), we use the retrieved paragraphs provided by Wang et al. (2017b) for these two datasets. To fine-tune our models for this task, the input token sequence is modified as *"<SEP>question </SEP>paragraph</SEP>"*. We apply linear layers over the last hidden features of our model to predict the start and end position of the answer span. We adopt two metrics including ExactMatch (EM) and loose $F_1$ (Ling & Weld, 2012) scores to evaluate our models.

**Baselines     BERT-FT$_{RACE+SWAG}$ (Huang et al., 2019)** is the BERT model sequentially fine-tuned on both RACE and SWAG datasets. **BiDAF (Seo et al., 2016)** adopts a bi-directional attention network. **AQA (Buck et al., 2018)** proposes to re-write questions and aggregate the answers generated by the re-written questions. **R^3 (Wang et al., 2017a)** is a reinforced model making use of a

ranker for selecting most confident paragraph. **Evidence Agg. (Wang et al., 2018)** proposes making use of the aggregated evidence from across multiple paragraphs to better determine the answer with re-rankers. **BERT (Xiong et al., 2020)** is the BERT re-implementation. **WKLM (Xiong et al., 2020)** is described in Section 2, which is adopted as the reader model to read multiple paragraphs to predict a single answer. **WKLM + Ranking (Xiong et al., 2020)** is a WKLM paragraph reader plus with a BERT based paragraph ranker to assign each paragraph a relevance score.

**Results and Discussion** The results on CosmosQA are shown in Table 3. Compared with BERT-FT$_{RACE+SWAG}$, our RoBERTa significantly achieves 11.89% improvement of accuracy. Compared to RoBERTa, K-ADAPTER (F+L) further improves the accuracy by 1.24%, which indicates that K-ADAPTER can obtain better commonsense inference ability. Moreover, the performance of ablated K-ADAPTER models, i.e., K-ADAPTER (F) and K-ADAPTER (L) are clearly better than RoBERTa, but slightly lose compared with RoBERTa+multitask. It is notable that K-ADAPTER (F+L) makes obvious improvement comparing with RoBERTa+multitask. This demonstrates that the combination of multiple knowledge-specific adapters could achieve better performance.

The results for open-domain QA are shown in Table 3. K-ADAPTER models achieve better results compared to other baselines. This indicates that K-ADAPTER models can make full use of the infused knowledge and accordingly benefit understanding the retrieved paragraphs to answer the question. Specifically, on SearchQA, K-ADAPTER (F+L) makes significant improvement of 4.01% $F_1$ scores, comparing with WKLM where the ranking scores are not used, and even has a slight improvement as compared to WKLM+Ranking. It is worth noting that K-ADAPTER models do not consider the confidence of each retrieved paragraph, while WKLM+Ranking utilizes ranking scores from a BERT based ranker. On the Quasar-T dataset, K-ADAPTER (F+L) also outperforms WKLM by 3.1% $F_1$ score and slightly outperforms WKLM+Ranking.

### 4.3 RELATION CLASSIFICATION

Relation classification aims to determine the correct relation between two entities in a given sentence. We adopt a large-scale relation classification dataset TACRED (Zhang et al., 2017).To fine-tune our models for this task, we modify the input token sequence by adding special token "@" before and after the first entity, adding "#" before and after the second entity. Then the token representations of the first special token "@" and "#" are concatenated to perform relation classification. We adopt micro $F_1$ score as the metric to represent the model performance as previous works.

**Baselines** C-GCN (Zhang et al., 2018) employs graph convolutional networks to model dependency trees. **BERT-large (Baldini Soares et al., 2019)** is a baseline BERT-large model. **BERT+MTB (Baldini Soares et al., 2019)** is a method of training relation representation without supervision from a knowledge base by matching the blanks. Other baseline models, such as ERNIE, KnowBERT, KEPLER and RoBERTa+multitask are described in Section 2 and 4.1.

**Results and Discussion** Table 4 shows the performances of different models on TACRED. The results indicate that K-ADAPTER models significantly outperform all baselines, which directly demonstrate our models can benefit relation classification. In particular, (1) K-ADAPTER models outperform RoBERTa, which proves the effectiveness of infusing knowledge into pre-trained model with adapters. (2) K-ADAPTER models gain more improvement compared with *RoBERTa+multitask*. This directly demonstrates injecting knowledge individually in K-ADAPTER way would help models make full use of knowledge.

Table 4: Results on the relation classification dataset TACRED.

| Model | P | R | $F_1$ |
|---|---|---|---|
| C-GCN (Zhang et al., 2018) | 69.90 | 63.30 | 66.40 |
| BERT-base (Zhang et al., 2019) | 67.23 | 64.81 | 66.00 |
| ERNIE (Zhang et al., 2019) | 69.97 | 66.08 | 67.97 |
| BERT-large (Baldini Soares et al., 2019) | - | - | 70.10 |
| BERT+MTB (Baldini Soares et al., 2019) | - | - | 71.50 |
| KnowBERT (Peters et al., 2019) | 71.60 | 71.40 | 71.50 |
| KEPLER (Wang et al., 2019) | 70.43 | 73.02 | 71.70 |
| RoBERTa | 70.17 | 72.36 | 71.25 |
| RoBERTa + multitask | 70.18 | 73.11 | 71.62 |
| K-ADAPTER (F+L) | 70.14 | 74.04 | **72.04** |
| K-ADAPTER (F) | 69.39 | 74.59 | 71.89 |
| K-ADAPTER (L) | 68.85 | 75.37 | 71.96 |

### 4.4 PROBING EXPERIMENTS

Although K-ADAPTER models have shown superior performance on knowledge-driven downstream tasks, it does not directly provide insights into whether our models infuse richer factual knowledge. Thus we utilize a LAMA (LAnguage Model Analysis) probe (Petroni et al., 2019) to examine the ability to memorize factual knowledge. Specifically, the LAMA probing task is under a zero-shot setting, which requires the language model to answer cloze-style questions about relational facts without fine-tuning, e.g., "Simon Bowman was born in [MASK]". The model needs to predict a distribution over a limited vocabulary to replace [MASK]. We report mean precision at one (P@1) macro-averaged over relations.

Table 5: P@1 on LAMA and LAMA-UHN across Google-RE and T-REx corpora.

| Corpus | Models | | | | | |
|---|---|---|---|---|---|---|
| | ELMo | ELMo5.5B | TransformerXL | BERT-large | RoBERTa$_{LARGE}$ | K-APDATER |
| LAMA-Google-RE | 2.2 | 3.1 | 1.8 | 12.1 | 4.8 | 7.0 |
| LAMA-UHN-Google-RE | 2.3 | 2.7 | 1.3 | 6.5 | 2.5 | 3.7 |
| LAMA-T-REx | 0.2 | 0.3 | 19.5 | 33.9 | 27.1 | 29.1 |
| LAMA-UHN-T-REx | 0.2 | 0.2 | 12.6 | 26.2 | 20.1 | 23.0 |

Table 6: Examples of generation for RoBERTa$_{LARGE}$ and K-ADAPTER. The last column reports the top ranked predicted tokens. Correct predictions are in **bold**.

| Query | Answer | Model | Generation |
|---|---|---|---|
| The native language of Mammootty is [MASK]. | Malayalam | RoBERTa | English, Tamil, Hindi, Sanskrit, Arabic, Chinese |
| | | K-ADAPTER | **Malayalam**, Tamil, Hindi, Mandarin, English |
| Ravens can [MASK]. | fly | RoBERTa | win, play, score, lose, run, drink, **fly**, roll, wait |
| | | K-ADAPTER | **fly**, swim, sing, shoot, kill, go, fish, drink, die |
| Sometimes virus causes [MASK]. | infection | RoBERTa | cancer, death, illness, blindness, paralysis |
| | | K-ADAPTER | cancer, illness, death, **infection**, disease |
| Sunshine Coast, British Columbia is located in [MASK]. | Canada | RoBERTa | Florida, California, Texas, Hawaii, Mexico |
| | | K-ADAPTER | **Canada**, Vancouver, Victoria, BC, Australia |
| iPod Touch is produced by [MASK]. | Apple | RoBERTa | **Apple**, Samsung, Qualcomm, LG, Microsoft |
| | | K-ADAPTER | **Apple**, HTC, Samsung, Motorola, Intel |

**Settings** We consider several language models including: ELMo (Peters et al., 2018), ELMo5.5B (Peters et al., 2018), Transformer-XL (Dai et al., 2019), BERT$_{LARGE}$ and RoBERTa$_{LARGE}$. We focus on LAMA-GoogleRE and LAMA-T-REx, which are aimed at factual knowledge. We also conduct probe experiments on LAMA-UHN (Poerner et al., 2019), a more "factual" subset of LAMA, by filtering out queries that are easy to answer from entity names alone. Different models have different vocabulary sizes. To conduct a more fair comparison experiment, we adopt the intersection of vocabularies and let every language model rank only tokens in this vocabulary following Petroni et al. (2019). For simplicity, we only compare K-APDATER (F) which is infused with factual knowledge, with other baseline models.

**Results and Discussion** Results are shown in Table 5. It is surprising that BERT$_{LARGE}$ performs better than RoBERTa$_{LARGE}$. There is one possible reason: BERT uses a character-level BPE (Gage, 1994) vocabulary, while RoBERTa considers byte-level BPE vocabulary. This finding indicates that, although using bytes makes it possible to learn a subword vocabulary that can encode any text without introducing "unknown" tokens, it might indirectly harm the model's ability to learn factual knowledge, e.g., some proper nouns may be divided into bytes. Thus in the following experiments, we do not take BERT into account.

K-ADAPTER outperforms other models (except for BERT) by a huge margin. As for LAMA, compared to RoBERTa$_{LARGE}$, K-ADAPTER obtains 2.2% and 1.2% P@1 improvement across Google-RE and T-REx, respectively. Moreover, compared to RoBERTa$_{LARGE}$, K-ADAPTER still achieves better results on LAMA-UHN. The results demonstrate that K-ADAPTER captures richer factual and commonsense knowledge than RoBERTa. Furthermore, Table 6 shows several examples for the generation of RoBERTa$_{LARGE}$ and K-ADAPTER for LAMA queries. From these examples, we can find that the objects predicted by K-ADAPTER are more accurate.

## 4.5 CASE STUDY

Table 7 gives a qualitative comparison example between K-ADAPTER and RoBERTa on relation classification dataset TACRED. The results show that, in most cases, the wrongly predicted logit value of RoBERTa and the logit value of the true label are actually quite close. For example, given "*New Fabris closed down June 16*", RoBERTa predicts "no_relation", but the true label "*city_of_birth*" ranks in second place. If a model could correctly predict the relationship between "*New Fabris*" and "*June 16*", then it needs to know that "*New Fabris*" is a company. Thanks to the factual knowledge in K-ADAPTER, it can help the model from predicting "no_relation" to predicting the correct category label.

Table 7: A case study for K-ADAPTER and RoBERTa on relation classification dataset TACRED. Underlines and wavy lines highlight the subject entities and object entities respectively. We report the top 3 ranked predictions.

| Input | True label | Model | Predicted label | Predicted logits |
|---|---|---|---|---|
| His former student Mark Devlin of the University of Pennsylvania was co-leader of the other , known as the Microwave Anisotropy Telescope . | schools_attended | K-Adapter | ['schools_attended', 'no_relation','founded'] | [12.67, 9.58, 5.26] |
| | | RoBERTa | ['no_relation', 'founded', "member_of"] | [9.18, 6.54, 5.07] |
| Graham had been in custody in Vancouver , British Columbia , since June . | cities_of_residence | K-Adapter | ['cities_of_residence', 'countries_of_residence', 'no_relation'] | [13.52,6.88,6.61] |
| | | RoBERTa | ['countries_of_residence', 'country_of_death', 'alternate_names'] | [7.14, 7.03, 6.83] |
| Vladimir Ladyzhenskiy of Russia died after she suffered a shock in the final of the spa world championship in Heinola , a southern city of Finland , on Saturday . | cause_of_death | K-Adapter | ['cause_of_death','origin','no_relation'] | [11.05, 7.65, 7.14] |
| | | RoBERTa | ['no_relation', 'cause_of_death', 'origin'] | [6.32, 5.90, 5.52] |
| You can't have a good season unless it starts well, " said Bill Martin, co-founder of ShopperTrak, on Saturday . | founded_by | K-Adapter | ['founded_by', 'member_of', 'employee_of'] | [10.25, 9.32, 7.37] |
| | | RoBERTa | ['no_relation', 'founded_by', 'employee_of'] | [10.01, 8.57, 5.42] |
| New Fabris closed down June 16 . | dissolved | K-Adapter | ['dissolved', 'no_relation', 'date_of_death'] | [12.94, 8.79, 6.83] |
| | | RoBERTa | ['no_relation', 'dissolved', 'date_of_birth'] | [11.44, 9.84, 3.31] |
| At Countrywide , which is finishing up a round of 12,000 job cuts, Chief Executive Angelo Mozilo said in announcing the Bank of America takeover last week that the housing and mortgage sectors were being strained " as never seen since the Great Depression . | dissolved | K-Adapter | ['dissolved', 'no_relation', 'date_of_death'] | [11.76, 6.89, 6.42] |
| | | RoBERTa | ['date_of_birth', 'date_of_death','no_relation'] | [7.44, 6.88, 6.33] |

## 5 CONCLUSION AND FUTURE WORK

In this paper, we propose a flexible and simple approach, called K-ADAPTER, to infuse knowledge into large pre-trained models. K-ADAPTER remains the original parameters of pre-trained models unchanged and supports continual knowledge infusion, i.e., new kinds of injected-knowledge will not affect the parameters learned for old knowledge. Specifically, factual knowledge and linguistic knowledge are infused into RoBERTa with two kinds of adapters, which are pre-trained on the relation classification task and dependency relation prediction task, respectively. Extensive experiments on three knowledge-driven downstream tasks demonstrate that the performance of each adapter achieves a significant improvement individually, and even more together. Probing experiments further suggest that K-ADAPTER captures richer factual and commonsense knowledge than RoBERTa. In future work, we will infuse more types of knowledge, and apply our framework to more pre-trained models.

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
