# OpenReview forum: "K-Adapter: Infusing Knowledge into Pre-Trained Models with Adapters"
_ICLR.cc/2021/Conference — Reject_

### Official Review · AnonReviewer4 · 2020-10-28
**As fine tuning a pre-trained model for specific tasks updates the original weights of that model and can lead to catastrophic forgetting, the authors propose to keep the pre-trained model's weights fixed and inject knowledge via separate adapter network that are pre-trained independently for specific tasks using internal representations of the pre-trained model.  The work learns two such adapters on top of RoBERTa and shows improved model performances on 3 different tasks using them.**

**Rating:** 6
**Confidence:** 3

**Review:**

##########################################################################
Reasons for score:

The authors propose a plug-in based adapter approach to allow for task specific parameter settings without updating the original pre-trained model which prevents the potential for catastrophic forgetting while also removing the need for separate models for separate tasks.  The work seems to build off Houlsby 19  as briefly cited, but its in plug-in nature seems easier to adopt for multiple tasks.  There is however not direct comparison with it or Cooper et al 19 ( https://arxiv.org/pdf/1902.02671.pdf ) which makes it difficult to assess.  The way in which the adaptors were pretrained was a little unclear to me.  The experiments are extensive and well done.


##########################################################################
Pros:
1) The number of experiments run ( 3 tasks on 6 datasets total ) are extensive and shows the K-adaptor approach can benefit from the factual adaptor in particular in giving better performance over RoBERTa ( with or without multi-task learning ).

2) The proposed adaptor seems concise and easily expanded to incorporate other knowledge sources ( though there are few details which could help clarify things see #2 in next section )

3) The probing task using LAMA to show how much factual knowledge has been memorized by the K Adaptor ( RoBERTA + facAdapter ) was well done and its discussion was very interesting.

##########################################################################
Cons:
1)  The proposed adapter solution is somewhat similar in nature to that of Houlsby 19  ( and to a lesser extent Cooper 19 ( https://arxiv.org/pdf/1902.02671.pdf ) ) and it feels like an omission to not discuss Houlsby 19 and make experimental comparisons against it discussing pros/cons more thoroughly especially since in the extensive experiments done in this work it is shown the linguistic adapter usually only adds a tenth of a percentage point when using RoBERTa with a single Factual adapter.  In this single adapter case then its not immediately evident how these models would differ and what the advantage is.   Both Houlsby and Cooper are evaluated on the GLUE benchmark and provide code.

2) I was a little confused as to how the adapters were specifically pre-trained and it might be a question of Figure 1b, but also sections 3.3 and 3.4 could have been expanded to clarify it a bit.  It is my understanding that when pre-training the facAdapter on the relation classification task for instance in Section 3.3, for a given example in T-REx-rc,  two entities and context are passed into RoBERTA whose weights remain fixed while those of the KIA units of the facAdapter are updated and the final hidden representations of RoBERTA and the facAdapter are concatenated to form an input representation of the entities given their context and this used for the actual task.  Is my understanding correct?  If so I'm confused as to how the subsequent pooling and concatenation actions are done.   Clarifying this process for 3.3 and 3.4 would be beneficial for clarity purposes and its not discussed in the supplemental materials either.

3) Your RoBERTa Large baseline already beats most of what you are comparing against which is fine as your adapters give gains ( again particularly the facAdapter), but it also would have been interesting to see what sort of gains would have been achieved using a different less powerful model as the base RoBERTa small or just plain BERT and additionally some sort of ablation testing or explanation on the choices made for the adapter networks themselves ( ie, N=2 Transformers etc , hidden layer size, etc ) though its possible this could be left for future work.  For clarity in Figure 2 where you show N x Transformer Layer ( and N=2), I'm assuming the first Transformer Layer feeds directly into the second Transformer Layer which then feeds into the Up Projection layer correct?  If so it might be better just to show two transformer layers like that instead and additionally, naming the projection layers Up and Down Projection Layer respectively.

##########################################################################

Questions during rebuttal period:

Please address and clarify the cons above

#########################################################################
Small typos:
In Abstract:  we propose K-ADAPTER, which remains the original parameters  .... "remains"  ->  "keeps" or "retains"

In Introduction: they fail to continual learning   .....  "fail at continual learning"
                              It remains the original representation    ....   "remains"  ->  "leaves"
                              (pg2) while remaining the original parameters of RoBERTa frozen...  "remaining"  ->   "keeping"

Section 3:     It remains the original representation    ....   "remains"  ->  "keeps"
              3.1:  Different from Houlsby et al. (2019) add adapter layers  ->  "In contrast to Houlsby et al. (2019) who add adapter layers"
              3.3:  all relations having lees than .... "lees" -> "less"

---

> ### Author Response · Authors · 2020-11-18
> **Reply to Review #4**
>
> Thanks for your thorough reading and positive comments, we have revised our paper accordingly. Please see our explanations to your questions below:
>
> A1: Our motivation is to support continual knowledge augmentation on pre-trained language model (LM). To achieve this, we develop a different adapter structure that works as outside plug-ins rather than internal adapters in Houlsby et al., (2019) and Cooper et al., (2019). This difference largely facilitates continual learning.
> In addition, the results presented by Houlsby et al., (2019) show that their model can achieve comparable performance (slightly loss) with fine-tuning pre-trained LM, but our model can achieve much better performance than fine-tuning pre-trained LM, and even better than multi-task fine-tuning pre-trained LM. Hence, we don’t conduct further experiments with those internal adapters.
> Moreover, our work aims to produce a pre-trained model that injects many kinds of knowledge. We have reported the results of our methods and many knowledge-enhanced baselines, including many recent ones. Both Houlsby et al., (2019) and Cooper et al., (2019) are not related to knowledge-enhanced works, so we believe it is not necessary to compare to them in downstream tasks.
>
> A2: Your understanding is almost right. One thing to point out, though, is that when pre-training the facAadapter on the relation classification task, instead of two entities and context being passed into RoBETa, only the text sentence is passed directly into RoBERTa (just like the usual way). Then the pooling operation is performed on the representations of the subject and object entities to get 2 entity representations respectively. Then, we concatenate two entity representations to perform relation classification. *To illustrate this process more clearly, we've revised the paper to add Figure 3 and Figure 4 in supplemental materials.* Figure 3 shows an overview of K-Adapter to inject specific knowledge by training a knowledge-specific adapter on the pre-training task. Figure 4 shows an example of using relation classification as a pre-training task to inject knowledge into K-Adapter.
>
> A3: (1) Yes, thanks for your advice. We study on RoBERTa as the community always wants to see improvements over a stronger baseline. We have conducted some experiments on less powerful baseline such as BERT, and the gain is larger. Meanwhile, we have released our project, so everyone can play with it with BERT, GPT or XLNet.
>
> (2) Thanks for your suggestion for adding some explanation or ablation test of how this network structure is chosen. There are mainly two reasons for we choose the current adapter structure. First, Jawahar et al., (2019) show that different layers of pre-trained models such as BERT learn different levels of semantic features, with surface features at the bottom, syntactic features in the middle and semantic features at the top. Meanwhile, we conduct preliminary experiments on ablation testing of adapter networks, and found that plugging the adapter into the bottom, middle and top layers of pre-trained LM respectively will bring larger gain.
> Reference: Ganesh Jawahar, Benoît Sagot, Djamé Seddah. What does BERT learn about the structure of language?. ACL 2019. We leave more detailed exploration as our future work.
>
> (3) Thanks for your helpful suggestion. We have updated Figure 2.
>
> To typos: Thanks for pointing typos. We have revised our paper.

---

### Official Review · AnonReviewer2 · 2020-10-28
**Modular knowledge injection**

**Rating:** 7
**Confidence:** 3

**Review:**

#### Summary

This submission proposes a general method (K-Adapter) for injecting knowledge (either factual or linguistic) into pre-trained language models.  The key architectural property of the approach is that K-Adapters are isolated from one another, allowing the use of multiple adapters without interference. These K-Adapter modules take hidden layer _inputs_ from the main pre-trained model (eg, BERT), and are pre-trained on their knowledge outputs before a fine-tuning phase where they feed into a joint downstream task-specific model along with the pre-trained model outputs.

#### Strong and weak points

The set of baselines seem strong, and the experimental results consistently show that using _either_ factual or linguistic knowledge K-Adapters improves, while using _both_ yields the best results.

The LAMA probing experiment is a nice sanity or validation test that the knowledge injection is achieving the desired effect. Being able to "hard-code" knowledge into the model in this way could be useful in a variety of applications. It is overselling it a bit to say the model captures "richer" commonsense knowledge, however.

The basic architectural idea is well-motivated and simple, in a good way.

The supplemental materials mostly provide additional reproducibility details on architectures, hardware used, learning rates, etc.

#### Recommendation (accept or reject) with one or two key reasons for this choice.

I recommend to accept. The proposed approach yields strong quantitative performance against solid and relevant baselines, and the LAMA experiments give some support to the hypothesis that it is doing so by capturing knowledge as intended. The general design pattern could spur further innovations in modular network designs or knowledge capture strategies as well.

#### Questions to clarify / additional evidence required

"BERT-MK inegrates fact triples from the knowledge graph." - how? I can follow the cite but this sentence provides little information.

"inject different types of knowledge independently" - is it correct to say then, that, by design, there can be no _beneficial_ interactions or synergies among different types of knowledge? Alternatively, in the fine-tuning phase, could different adapters interact or affect each other via the downstream coupling in the task-specific layers? Is this observed in practice?

How should the reader think about the relative magnitude of the presented improvements? At one point I see "K-ADAPTER (F+L) makes significant improvement of ..." but I believe "significance" is only meant coloquially here.

Section 3.1: how was this structure chosen, what was the motivation or intuition here?

What limits, if any, do you foresee with the use of separate parallel "knowledge modules" like this? Could we use 10, 100, 1000 K-Adapters?

#### Additional feedback to improve

It would be helpful to cite Ling and Weld 2012 (or similar) for the definition of "loose" micro/macro F1, or briefly explain it inline in the evaluation setup.  Likewise for the "catastrophic forgetting" phenomenon affecting other knowledge injection attempts - is there some previous work explicitly demonstrating this problem when using multiple knowledge sources? If not, it would have been interesting to have an experiment of this sort in this work.

---

> ### Author Response · Authors · 2020-11-18
> **Reply to Review #2**
>
> Thank you for your thorough reading, insightful comments, and valuable suggestions, we have updated the paper accordingly. Below we address the concerns mentioned in the review:
>
> Q1: "BERT-MK integrates fact triples from the knowledge graph." - how? I can follow the cite but this sentence provides little information.
>
> A1: Due to page limits at the time of submission of the first draft, we didn't present related works in detail. We have updated the content as follows: BERT-MK integrates fact triples from the knowledge graph. For each entity, it samples incoming and outcoming instances from the neighbors on the knowledge graph, and replaces the head or tail entity to create negative instances. The model is learned to discriminate between real and fake facts.
>
>
> Q2: "inject different types of knowledge independently" - is it correct to say then, that, by design, there can be no beneficial interactions or synergies among different types of knowledge?
>
> A2: "inject different types of knowledge independently" is in the pre-training phase where is no information flow between different adapters, thus different adapters are efficiently trained in a distributed way. We use separate adapters to support continual knowledge augmentation and address the problem of catastrophic forgetting. We can inject each kind of knowledge into the corresponding knowledge-specific adapter.
>
>
> Q3: In the fine-tuning phase, could different adapters interact or affect each other via the downstream coupling in the task-specific layers? Is this observed in practice?
>
> A3: In the fine-tuning phase, different adapters could interact or affect each other in the feature fusion layer and task-specific layers due to backward propagation. Leveraging different kinds of knowledge is helpful for downstream tasks. Because results demonstrate that each adapter improves the performance, and the combination of both adapters always brings further improvements.
>
>
> Q4: How should the reader think about the relative magnitude of the presented improvements? At one point I see "K-ADAPTER (F+L) makes significant improvement of ..." but I believe "significance" is only meant coloquially here.
>
> A4: "Significant improvement" here means that "improves by a large margin". On SearchQA dataset, K-Adapter (F+L) outperforms WKLM by 4.01% F_1 scores (as shown in Table 3), which is regarded as a large improvement. Thanks for pointing out, we consider changing this expression.
>
>
> Q5: How was this adapter structure chosen, what was the motivation or intuition here?
>
> A5: Please allow us to explain it to you in two ways:
> - (1) If your question is why we design the adapter works as outside plug-ins, our motivation is to support continual knowledge augmentation on the pre-trained language model (LM). To achieve this, we develop a different adapter structure that works as outside plug-ins rather than internal adapters in Houlsby et al., (2019). This difference largely facilitates continual learning. In addition, the results presented by Houlsby et al., (2019) show that their model can achieve comparable performance (slightly loss) with fine-tuning pre-trained LM, but our model can achieve much better performance than fine-tuning pre-trained LM, and even better than multi-task fine-tuning pre-trained LM.
> - (2) If your question is how we choose the structure such as where the adapter layers plug into RoBERTa layers, there are mainly two reasons. First, Jawahar et al., (2019) show that different layers of pre-trained models such as BERT learn different levels of semantic features, with surface features at the bottom, syntactic features in the middle and semantic features at the top. Meanwhile, we conduct preliminary experiments on ablation testing of adapter networks, and found that plugging the adapter into the bottom, middle and top layers of pre-trained LM respectively will bring larger gain.
>
> Reference: Ganesh Jawahar, Benoît Sagot, Djamé Seddah. What does BERT learn about the structure of language?. ACL 2019
>
>
> Q6: Do you foresee with the use of separate parallel "knowledge modules" like this? Could we use 10, 100, 1000 K-Adapters?
>
> A6: Ideally, K-Adapter can support a large number of adapters. This is our motivation to design K-Adapter to support continual knowledge augmentation on pre-trained language model (LM) and don’t suffer from catastrophic forgetting. In this paper, we conduct experiments with two adapters. We will introduce more adapters in future work.
>
>
> Q7: Additional feedback to improve.
>
> A7: Thanks for your helpful suggestion, we have added citations in our revised draft accordingly.

---

### Official Review · AnonReviewer1 · 2020-10-30
**Interesting work but limited by analysis and applicability**

**Rating:** 4
**Confidence:** 4

**Review:**

Summary:
The paper proposes a novel approach of incorporating different types of world knowledge sources contained in texts such as facts or linguistic syntax. To do this, they introduce additional transformers layers between the layers of a pre-trained language model such as Roberta and term this model as "K-Adapters", where the K stands for K different streams of knowledge.

Pros:
- Incorporating different sources of information into a pre-trained model such as Roberta is an interesting idea.
- The proposed approach is simple and interesting and it scales to many different types of information as the different adapters can be trained in parallel with the weights of the pre-trained LM being fixed.
- Performance gains on different classification tasks such as entity tying, question-answering, and relation classification highlights the utility of the approach.

Cons:
- Sec 1: Introduction: In the introduction, there are multiple mentions of the phrase “rich knowledge” but it is unclear what do the authors mean by that in the context of pre-trained language models. Some recent works such as “https://arxiv.org/abs/2002.08910, https://arxiv.org/abs/1909.01066 ” suggest that pretrained language models do indeed contain a lot of world factual knowledge. Hence, the statement in the paper that pertained LMs lack world knowledge is contradicting these works.
- There is also a frequent mention of catastrophic forgetting of knowledge during the finetuning step. I tend to disagree that this is necessarily bad for a pretrained model, because it has been shown that finetuning pre-trained LMs perform well in open-domain question answering where some degree of world knowledge is needed.
- Furthermore, producing entangled representations may not necessarily be a negative thing, if multi-task learning approaches are able to show an increase in performance due to knowledge injection.
- In Table 1, dependency parser doesn't really fall under the same class of knowledge sources such as Wordnet or Wikidata. A dependency parser may be able to provide some sort of syntactic structure of the underlying text. Moreover, such syntactic information is not always generalizable to different domains and thus has the limitation of not being accurate enough.
- The Introduction section is not well-motivated and does not present convincing arguments as to why external knowledge infusion is really required in some tasks. It just states that knowledge infusion using k-adapter model outperforms Roberta models in different tasks.
- In Section 3.1, not enough space has been allocated to explain the Adapter model in detail. If the authors had used mathematical notation or equations for explanation, then it would have been much more clear.
- In Section 4, it is mentioned that they select three downstream tasks for evaluating their models. However, the paper doesn't provide justifications as to why these tasks were selected, how can these tasks highlight the importance of k-adapter model, etc.
- In the results table 2 and table 4, as the performance improvements are somewhat marginal, it is important to know if these improvements are statistically significant or not. The paper doesn't report if the results are from single run or the mean of multiple runs.
- I have concerns about data leakage during the pre-training step. As the factual adapter makes use of supervised relation classification dataset (T-REx), I feel that there might be some overlap between entity typing and relation classification datasets used for evaluating the performance of the model. The authors should present an analysis as to what degree of overlap if any is present during the pre-training and evaluation tasks.
- The paper lacks a detailed analysis section that could explain as to which test examples are getting correctly classified when using k-adapter model in tasks like relation classification, entity typing compared to other baseline approaches such as Roberta, Roberta + multitask. Currently, the paper pays just too much emphasis on raw numbers or performance improvements in various tasks.
- The results of the probing experiments suggests that BERT-large model vastly outperforms k-adapter model on Google-Re and T-REx datasets probing datasets. This raises an important question over the validity of the results in different downstream tasks. For a fair comparison with baselines, the authors should compare the performance of the k-adapter model with BERT-large + multitask across different tasks.
- In almost all of the experiments, the authors use Roberta as the underlying pre-trained language model. For demonstrating generalization to different pre-trained LMs, the paper should also evaluate when k-adapter model is trained when BERT-large or T5-large are used as underlying models in place of Roberta.

Grammar errors:
- page 1, 3rd line from bottom: remains -> retains
- section 3.3, information that concerned -> information that is concerned
- section 3.4, father index is commonly referred to as head index of a word.

---

> ### Author Response · Authors · 2020-11-18
> **Reply to Review #1**
>
> Thank you for your detailed review. We have revised our paper accordingly, especially for adding experimental analysis and clarifying model structure. We address specific questions below.
>
> Q1: The statement in the paper that pertained LMs lack world knowledge is contradicting these works.
>
> A1: Pre-trained language models (LM) and knowledge infusion into pre-trained LMs are perpendicular, not contradictory. As you mentioned two papers suggest that BERT contains a lot of factual knowledge, but there are other works such as “Poerner et al. (2019) https://arxiv.org/pdf/1911.03681v1.pdf” suggest that the impressive performance of BERT is partly due to reasoning about entity names. Beside, Kassner & Schütze (2019) observe that pre-trained LMs are equally prone to generate facts (“birds can fly”) and their negation (“birds cannot fly”), thus this casts doubt on the claim have adequately learned factual knowledge. These observations motivate us and other researchers to study the injection of knowledge into pre-trained models. As stated in Related Work and Table 1, there is a growing interest in enhancing pre-trained language models with knowledge. For instance, ERINE (Zhang et al., 2019), KnowBERT (Peters et al., 2019), WKLM (Xiong et al., 2020) inject entity information to capture better factual knowledge.
>
> Reference:
> - [1] Nina Pörner, Ulli Waltinger, Hinrich Schütze:BERT is Not a Knowledge Base (Yet): Factual Knowledge vs. Name-Based Reasoning in Unsupervised QA. CoRR abs/1911.03681 (2019)
> - [2] Nora Kassner, Hinrich Schütze: Negated LAMA: Birds cannot fly. In ACL, 2020
>
> Q2: There is a frequent mention of catastrophic forgetting of knowledge during the finetuning step.
>
> A2: In this paper, we are talking about the catastrophic forgetting during the pre-training step, not the fine-tuning step. For instance, we have tried to inject two kinds of knowledge. As time goes on, you may want to integrate a lot of other knowledge (e.g., 10, 100, 1000) in the future. We want to introduce a method or a solution that supports continual knowledge injection and without suffering from catastrophic forgetting.
>
> Q3: Producing entangled representations may not necessarily be a negative thing, if multi-task learning approaches are able to show an increase in performance due to knowledge injection.
>
> A3: Compared with RoBERTa_multitask, K-Adapter makes major improvements on serval datasets, such as FIGER and Quasar-T. In addition, compared with RoBERTa_multitask baseline, the greatest advantage of K-Adapter is flexibility and expandability, which can support continual knowledge injection. The multitask methods do not support continual learning. If there is one new knowledge to be injected, the multitask model needs to be entirely retrained.
>
> Q4: About dependency parser in Table 1
>
> A4: The dependency parser in Table 1 means linguistic knowledge from dependency parsing. Linguistic knowledge is implicitly contained in natural language texts, e.g., syntactic and semantic information. In this work, we acquire linguistic knowledge from dependency relationships among words in natural language text. We run the off-the-shell dependency parser on a part of Book Corpus to obtain this pre-training dataset.
>
> Q5: More explanation of the Adapter model in detail would have much more clear.
>
> A5: Thanks for your suggestion. To illustrate our K-Adapter more clearly, we've revised the paper to add Figure 3, 4, 5 in supplemental materials. Figure 3 shows an overview of K-Adapter to inject specific knowledge by training a knowledge-specific adapter on the pre-training task. Figure 4 shows an example of using relation classification as a pre-training task to inject knowledge into K-Adapter. Figure 5 illustrates that fine-tuning K-Adapter just like what the original RoBERTa does. We hope those figures will give readers a better understanding of K-Adapter.
>
> Q6: Why select these three downstream tasks for evaluation.
>
> A6: We follow the knowledge-driven datasets used in previous works for evaluation. For instance, OpenEntity and Quasar-T are entity-centroid tasks that require specific factual knowledge. Achieving superior performance on these knowledge-driven downstream tasks has proven our model can capture knowledge to enhance the pre-trained model. But it does not directly provide insights into whether our models infuse richer factual knowledge. Thus we further utilize a LAMA probe to examine the ability to memorize factual knowledge.
>
> Q7: The performance improvements are somewhat marginal in Table 2,4.
>
> A7: (1) In Table 2, K-Adapter (F+L) improves 1.38% F1 over RoBERTa on OpenEntity. K-Adapter (F+L) improves 3.54% and 2.71% micro F1 over WKLM and RoBERTa respectively. In Table 4, K-Adapter (F+L) outperforms RoBERTa by 0.79% F1 and KEPLER by 0.34% F1. Those can be regarded as large improvements on these datasets. (2) As illustrated in the supplemental material, for all experiments, we set the random seed to be 42 for reproducibility.

---

> ### Author Response · Authors · 2020-11-18
> **Reply to Review #1 (continued)**
>
> Q8: Have concerns about data leakage during the pre-training step because of using T-Rex.
>
> A8: Yes, we agree with you that might occur data leakage during pre-training step with T-Rex. We have taken this into account before and have excluded the data that is relevant in the test set from the evaluation datasets before we conduct experiments.
>
> Q9: Lacks a detailed analysis that could explain as to which test examples are getting correctly classified when using K-Adapter.
>
> A9: Thanks for your advice. We add a case study in our paper as Section 4.5, where Table 6 gives a qualitative comparison example between K-Adapter and RoBERTa on relation classification dataset TACRED. The results show that, in most cases, the wrongly predicted logit value of RoBERTa and the logit value of the true label are actually quite close. For example, given "New Fabris closed down June 16", RoBERTa predicts "no_relation", but the true label "city_of_birth" ranks in second place. If a model could correctly predict the relationship between "New Fabris" and "June 16", then it needs to know that "New Fabris" is a company. Thanks to the factual knowledge in K-Adapter, it can help the model from predicting "no_relation" to predicting the correct category label.
>
> Q10: The results of the probing experiments suggests that BERT-large model vastly outperforms k-adapter model, the authors should compare the performance of the k-adapter model with BERT-large + multitask across different tasks.
>
> A10: Please allow us to explain it to you in two aspects.
> - (1) As we illustrated in our paper, BERT performs better than RoBERTa on probing experiments. Sun et al., (2020) observe the same phenomenon in their paper. We conjecture that it is mainly because BERT uses a character-level BPE vocabulary, while RoBERTa uses a larger byte-level BPE vocabulary. This implies that dividing words into finer sub-words indirectly harms the ability of models with byte-level BPE to learn knowledge. Nonetheless, K-Adapter also outperforms its baseline, RoBERTa, by a large margin. For instance, K-Adapter improves over RoBERTa by 2% and 2.9% on LAMA-T-REx and LAMA-UHN-T-REx respectively. The results demonstrate that K-Adapter can capture richer factual knowledge than RoBERTa.
> Reference: Tianxiang Sun, Yunfan Shao, Xipeng Qiu, Qipeng Guo, Yaru Hu, Xuanjing Huang, Zheng Zhang: CoLAKE: Contextualized Language and Knowledge Embedding. CoRR abs/2010.00309 (2020)
> - (2) In this paper, we have compared K-Adapter to RoBERTa and RoBERTa+multitask. The results show that K-Adapter outperforms both RoBERTa and RoBERTa+multitask. RoBERTa+multitask is a stronger baseline than BERT+multitask, so we think it is not necessary to compare K-Adapter to BERT+multitask.
>
> Q11: The paper should also evaluate when K-Adapter model is trained when BERT-large or T5-large.
>
> A11: We study on RoBERTa as the community always wants to see improvements over a stronger baseline. We have conducted some experiments on BERT-large, and the gain is larger. However, T5-large is much larger than BERT-large and RoBERTa-large, we are unable to reproduce a T5-large based K-Adapter due to resource constraints. We are happy to leave those experiments as our future work. In addition, we have released our code, so everyone could play with it with BERT, GPT, XLNet or even T5.
>
> Q12: About some typos.
>
> A12: Thanks for pointing typos. We have revised our paper.

---

### Official Review · AnonReviewer3 · 2020-10-30
**A good paper in general**

**Rating:** 6
**Confidence:** 4

**Review:**

The paper proposes a new approach to inject knowledge into pre-trained language representation models (PLMs). Instead of tuning the original PLM parameters, the paper plugs in new adapters for knowledge injection to avoid catastrophic forgetting.

Pros:
* Injecting knowledge into PLMs is an advanced topic. The authors focus on the catastrophic forgetting problem during knowledge injection.
* Evaluation is solid. The authors evaluate their model on three downstream tasks and show that the adapters improve the performance.
* The paper is well written and can be easily understood.

Cons:
* The approach is simple but achieves good performance over a variety of tasks. I appreciate that the authors conduct the knowledge probing experiment but its P@1 is quite low and worse than BERT. Some more explanations are expected.

---

> ### Author Response · Authors · 2020-11-18
> **Reply to Review #3**
>
> Thanks for your thorough reading and positive comments! We hope to be able to address your concern below.
>
> As illustrated in our paper, BERT performs much better than RoBERTa on probing experiments. Sun et al., (2020) observe the same phenomenon in their paper. We conjecture that it is mainly because BERT uses a character-level BPE vocabulary, while RoBERTa uses a larger byte-level BPE vocabulary. This implies that dividing words into finer sub-words indirectly harms the ability of models with byte-level BPE to learn knowledge. Though, K-Adapter outperforms its baseline, RoBERTa, by a large margin. For instance, K-Adapter improves over RoBERTa by 2% and 2.9% on LAMA-T-REx and LAMA-UHN-T-REx respectively. The results demonstrate that K-Adapter can capture richer factual knowledge than RoBERTa.
>
> Reference: Tianxiang Sun, Yunfan Shao, Xipeng Qiu, Qipeng Guo, Yaru Hu, Xuanjing Huang, Zheng Zhang: CoLAKE: Contextualized Language and Knowledge Embedding. CoRR abs/2010.00309 (2020)

---

### Public Comment · ~Zaiqiao_Meng1 · 2020-11-10
**Request for codes**

Hi, thanks for this interesting work. Especially the 'outside plug-ins' adapters really inspired me for my current work. May I ask you if the codes of your model can be publicly available to reproduce the results?

---

> ### Author Response · Authors · 2020-11-17
> **Codes have been publicly available**
>
> Thank you so much for your recognition of this work. Our project has been released.

---

### Decision · Program_Chairs · 2021-01-07
**Final Decision**

**Decision:**

Reject

**Comment:**

The paper augments pre-trained language models by introducing “adapter”, where each adapter is another language model pre-trained for a specific knowledge source (e.g., Wikidata) and an objective (e.g., relation classification). The representation from each adapter is concatenated to the representation from the generic LM. Specifically, they introduce two adaptors, “factual” (mostly derived from Wikipedia), and “linguistic” (from dependency parser), and the experiment shows modest improvements over various benchmarks.

This is a borderline paper, as both methods and experiments are reasonable yet not very novel or strong. The clarity of the paper can be improved (as pointed by R1 and R4), without any mathematical notations, model details are to be interpolated from figures. The novelty is limited and experimental rigor can be improved (i.e., for many settings, gains are fairly small and no variance reported).